

# PotatoG-DKB: a potato gene-disease knowledge base mined from biological literature

Congjiao Xie[1,2], Jing Gao[1,2,3], Junjie Chen[1,2] and Xuyang Zhao[1]

[1] College of Computer and Information Engineering, Inner Mongolia Agricultural University, Hohhot, Inner Mongolia, China
[2] Inner Mongolia Autonomous Region Key Laboratory of Big Data Research and Application for Agriculture and Animal Husbandry, Hohhot, Inner Mongolia, China
[3] Inner Mongolia Autonomous Region Government Service and Data Management Bureau, Hohhot, Inner Mongolia, China

## ABSTRACT

**Background:** Potato is the fourth largest food crop in the world, but potato cultivation faces serious threats from various diseases and pests. Despite significant advancements in research on potato disease resistance, these findings are scattered across numerous publications. For researchers, obtaining relevant knowledge by reading and organizing a large body of literature is a time-consuming and labor-intensive process. Therefore, systematically extracting and organizing the relationships between potato genes and diseases from the literature to establish a potato gene-disease knowledge base is particularly important. Unfortunately, there is currently no such gene-disease knowledge base available.

**Methods:** In this study, we constructed a Potato Gene-Disease Knowledge Base (PotatoG-DKB) using natural language processing techniques and large language models. We used PubMed as the data source and obtained 2,906 article abstracts related to potato biology, extracted entities and relationships between potato genes and related disease, and stored them in a Neo4j database. Using web technology, we also constructed the Potato Gene-Disease Knowledge Portal (PotatoG-DKP), an interactive visualization platform.

**Results:** PotatoG-DKB encompasses 22 entity types (such as genes, diseases, species, *etc.*) of 5,206 nodes and 9,443 edges between entities (for example, gene-disease, pathogen-disease, *etc.*). PotatoG-DKP can intuitively display associative relationships extracted from literature and is a powerful assistant for potato biologists and breeders to understand potato pathogenesis and disease resistance. More details about PotatoG-DKP can be obtained at https://www.potatogd.com.cn/.

# INTRODUCTION

Potato (*Solanum tuberosum* L.) is a member of the Solanaceae family. Because of its high nutritional and economic value, potato is the fourth largest food crop worldwide, following wheat, maize, and rice (*Kondhare, Natarajan & Banerjee, 2020*). In 2022, the Food and Agriculture Organization of the United Nations (FAO) reported that global potato

Corresponding author
Jing Gao, gaojing@imau.edu.cn

production surpassed 470 million tons, and as a major potato producing country, China's potato production exceeded 190 million tons (*FAO, 2022*). However, potatoes suffer from various diseases and pest infestations, seriously impairing tuber quality and yield (*Yuen, 2021*; *Enciso-Rodriguez et al., 2018*). Fungicides currently serve as an essential shield against diseases and pests (*Hernández, 2014*; *Tiwari et al., 2021*); unfortunately, their pervasive application poses both environmental and health risks (*Majeed et al., 2017a, 2017b*). As crop genomes continue to emerge, an increasing number of biologists are investigating gene-disease relationships at the gene level and publishing their findings.

A knowledge base facilitates the integration and efficient retrieval of information. Currently, there are three main methods for constructing a knowledge base: one based on genomic sequences, another based on ontology, and the third based on literature mining. Genomic sequencing provides detailed information on the genetic sequences of organisms, enabling a comprehensive understanding of gene structure and function. However, a knowledge base built with genomic sequencing lacks correlations between findings. The ontology-based approach typically relies on predefined ontologies, which include formalized descriptions of concepts, properties, and relationships. In contrast, literature mining extracts key information from a vast amount of scientific literature, establishing relationships between different pieces of knowledge. To date, the PubMed database houses more than 12,787 scientific articles related to *Solanum tuberosum*. A potato-related knowledge network based on literature mining can aid researchers in quickly finding relevant studies and data, thereby promoting the discovery of new biological mechanisms.

Multiple potato knowledge bases have been created, as shown in Table 1. The Spud DB (*Hirsch et al., 2014*), based on genome sequences and associated annotation datasets, offers phenotypic and genotypic data, functional annotation, gene ontology (GO), and BLAST databases from 250 potato clones. The SolRgene (*Vleeshouwers et al., 2011*) contains the R gene and R gene homologues of *Solanum* petota; the resistance of *Solanum* petota to late blight was analyzed through high-throughput disease experiments and field tests under various laboratory conditions. The PoMaMo database (*Meyer, Nagel & Gebhardt, 2005*) is a comprehensive archive of potato genome data. *Choe et al. (2018)* developed a comparative analysis tool and a genome viewer using web technology to identify homologous genes based on collinearity and sequence similarity among tomato, pepper, and potato species. Numerous plant genome databases exist that are not specific to potato, such as Ensembl plants (*Bolser et al., 2017*) with around 70 plant genomes, Crop Ontologies (*Shrestha et al., 2012*), which provides ontology-based descriptors for crop traits and standard variables for over 20 crops, and the Planteome database (*Cooper et al., 2018*), which provides comprehensive resources for reference ontology, plant genomics, and phenotypic genomics. Some knowledge bases also use the literature mining method, including the *Solanum tuberosum* knowledge base (*Ivanisenko et al., 2018*), which is a knowledge base of molecular genetic regulation and metabolic pathways. *Singh et al. (2021)* took the flesh color of potatoes as the main trait and trained an NLP model to recognize relevant biological entities and relationships (genes, proteins, metabolites, and traits) in a manually annotated *corpus* of 34 full-text potato articles. The International Potato Centre

**Table 1 Potato-related knowledge bases.** For each potato-related knowledge base introduced, this table includes the name of the knowledge base, the methods used for its construction, the species covered by the knowledge base, the primary data content stored in the knowledge base, the scale of the knowledge base, and the URL.

| Knowledge base | Method | Species | Content | Scale | Source URL |
|---|---|---|---|---|---|
| Spud DB (*Hirsch et al., 2014*) | Genomic sequence | Potato | Potato genome sequence and associated annotation datasets | 250 potato clones | http://spuddb.uga.edu/ |
| SolRgene (*Vleeshouwers et al., 2011*) | Genomic sequence | Solanum | R genes to *Phytophthora infestans* | 1,062 accessions, and 12,584 genotypes | https://www.plantbreeding.wur.nl/SolRgenes/ |
| PoMaMo (*Meyer, Nagel & Gebhardt, 2005*) | Genomic sequence | Potato | Potato genome data | Sequence data and molecular maps of all 12 potato chromosomes with about 1,000 mapped elements | https://www.gabipd.org/projects/Pomamo/ |
| Ensembl plants (*Bolser et al., 2017*) | Genomic sequence | Plant | Genome data | 39 sequenced plant species | https://plants.ensembl.org/index.html |
| Crop ontology (*Shrestha et al., 2012*) | Ontology | Plant | Validated trait names, annotation of phenotypic and genotypic data | Eight specific crop ontologies and trait dictionaries (cassava, chickpea, common bean, groundnut, maize, Musa, potato, rice, sorghum, and wheat). | http://www.cropontology.org/ |
| Planteome (*Cooper et al., 2018*) | Ontology | Plant | Reference ontologies, plant genomics and phenomics | 95 plant taxa | http://www.planteome.org |
| *Solanum tuberosum* knowledge base (*Ivanisenko et al., 2018*) | Literature mining | Potato, Maize, Rice, Arabidopsis thaliana | Molecular genetic regulation and metabolic pathways | 9,000 full-text articles and more than 130,000 abstracts from PubMed | http://www-bionet.sysbio.cytogen.ru/and/plant/ |
| (*Singh et al., 2021*) | Literature mining | Potato, Tomato, Eggplant, Capsicum | Flesh color phenotypic trait knowledge network | 4,023 PubMed abstracts of plant genetics-based articles | |
| CIP (*Mihovilovich et al., 2015*) | | Potato | Germplasm bank and Gene Resource Bank | | https://cipotato.org/genebankcip/ |
| AHDB | | Potato | Various aspects of potato production, including seed health, agronomy, crop protection, storage, and crop quality | | https://potatoes.ahdb.org.uk/ |

(CIP; *Mihovilovich et al. (2015)*) and the UK Agriculture and Horticulture Development Board (AHDB) also provide abundant potato germplasm resources. As shown in Table 1, existing knowledge bases constructed through literature mining do not include a potato gene-disease knowledge base.

To provide the associations between potato genes and diseases, we used natural language processing (NLP) technology and large language models (LLM) to construct the Potato Gene-Disease Knowledge Base.

There are three primary methods of extracting knowledge from scientific literature: manual extraction, rule-based extraction, and machine learning. The manual method, although highly accurate, is extremely time-consuming (*Xing et al., 2018*). Rule-based and template-based methods heavily rely on biological experts, as illustrated by *Choi et al. (2016)*, *Larmande, Do & Wang (2019)*, *Saik et al. (2017)*. The traditional machine learning approach heavily relies on manually annotated training data (*Mausam, 2016*; *Fiorini et al., 2018*). Methods based on LLM are increasingly being used and have achieved state-of-the-art results in various fields (*Wadhwa, Amir & Wallace, 2023*; *Sun et al., 2024*).

This article introduces the PotatoG-DKB, which was constructed using literature mining and natural language processing technology to extract biological entities such as genes, diseases, fungi, bacteria, and pests. LLM was used to determine the semantic relationships between these entities. These relationships were then mapped onto a Potato Gene-Disease Knowledge Graph (PotatoG-DKG). This multi-relational graph includes nodes that represent biological entities and edges that signify relationships between these entities. The PotatoG-DKG serves as a comprehensive knowledge base that integrates diverse information and displays correlations in a graph structure. This format simplifies access to relevant knowledge and can support research efforts. We also developed a visualization platform for PotatoG-DKB using web technology. The Potato Gene-Disease Knowledge Portal (PotatoG-DKP) allows biologists and potato breeders to intuitively explore information embedded in the literature.

## MATERIALS AND METHODS

In this study, scientific literature served as the primary source of experimental data. To construct the PotatoG-DKB, unstructured information was extracted from potato-related scientific literature using a range of techniques. First, the query "potato and gene and disease" was used on PubMed (https://pubmed.ncbi.nlm.nih.gov/) to acquire 2,906 literature abstracts. The PubTator tool was then used to extract the basic biological entities, such as genes, proteins, species, and mutations, from these abstracts. Other biological entities were extracted using a pattern-matching approach. To improve efficiency, the text of the original abstracts was broken into sentences, and sentences containing at least two biological entities were retained. A large language model (LLM) with constructed prompts was used to extract data on potato gene-disease relationships and other related information. To ensure data accuracy, the obtained data was manually curated into PotatoG-DKB. The structured data was stored in a Neo4j graph database. To enable potato researchers and breeders to quickly access relevant knowledge, we developed the PotatoG-DKP. The construction process of the PotatoG-DKP is illustrated in Fig. 1.

### Preprocessing

The data sources analyzed in this article were retrieved from PubMed (https://pubmed.ncbi.nlm.nih.gov/), a search engine maintained by the National Center for Biotechnology

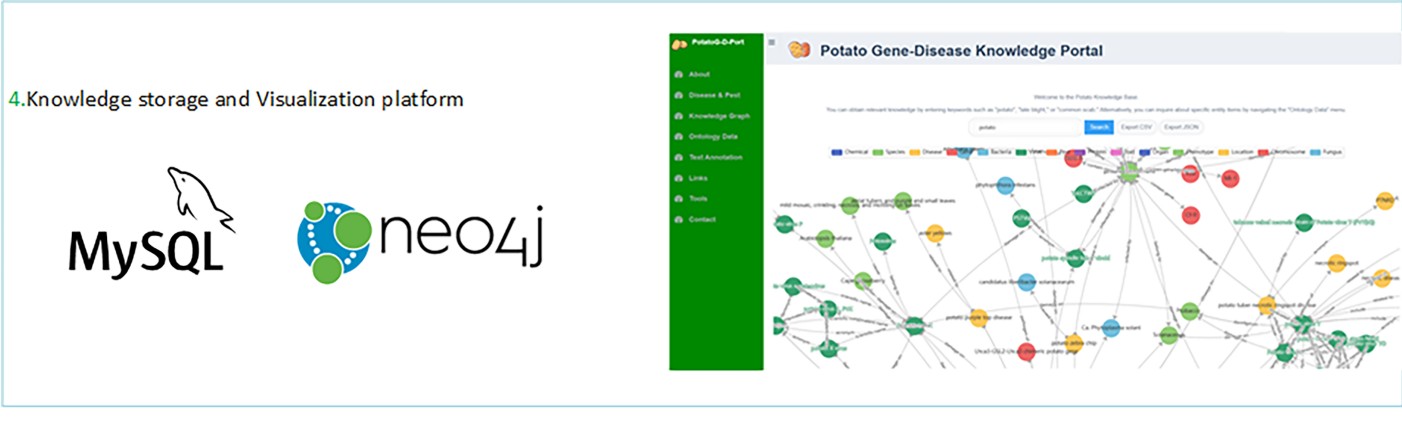

4.Knowledge storage and Visualization platform

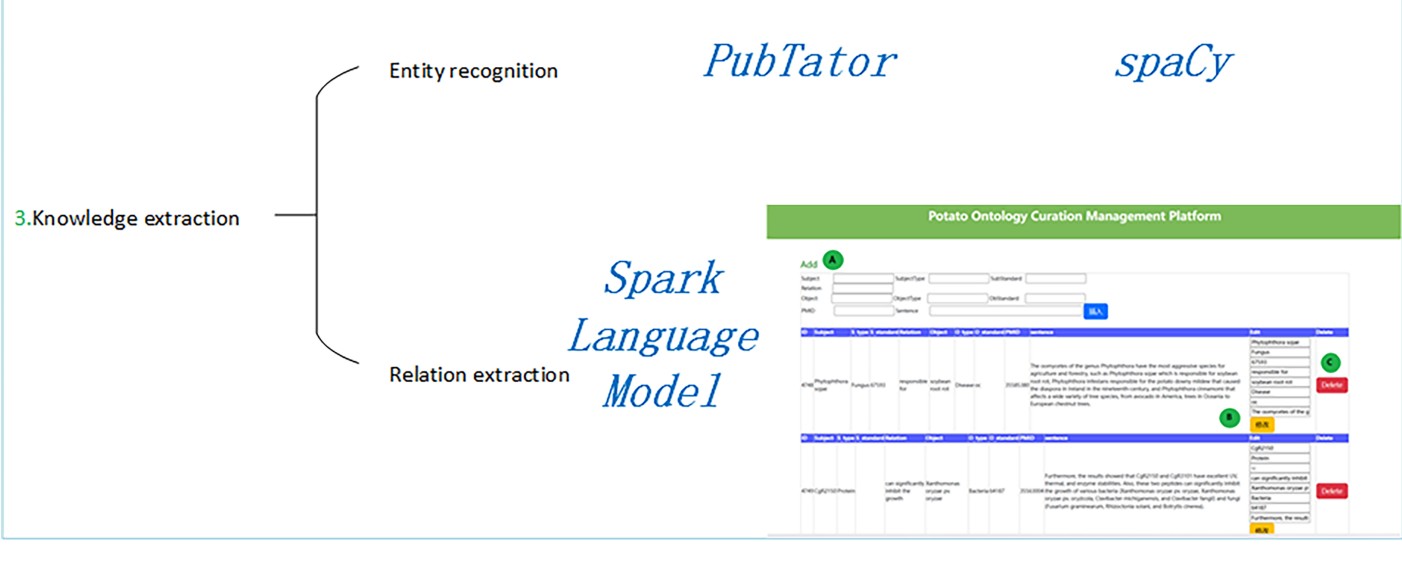

3.Knowledge extraction

Entity recognition

*PubTator*  *spaCy*

*Spark Language Model*

Relation extraction

2. Preprocessing

1. Potato gene-disease data source from NCBI

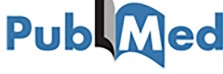

**Figure 1** **The flowchart for constructing the potato gene-disease knowledge base.** Txt icon designed by iconixar on Flaticon.com. Coding icon designed by orvipixel on Flaticon.com.

Information (NCBI) at the National Library of Medicine (NLM; *Wei et al. (2019)*). For this study, "potato and gene and disease" was used as the search string in PubMed, and 2,906 final abstracts were selected from the search results. All sources were saved in the PubMed_ID (PMID) format. PubTator Central provides automatic annotations of biological concepts from PubMed abstracts. The PubTator API (https://www.ncbi.nlm.nih. gov/research/pubtator/api.html) provides source code downloads in Perl, Python, and

Java, and the annotation formats include PubTator, biocxml, and biocjson. PubTator was used to annotate the obtained sources in PubTator format, including titles, abstracts, and annotations of biological entities, saved in ".txt" format. We used rule-based syntax in sentence segmentation to avoid inaccuracies associated with punctuation-based splitting (https://docs.python.org/3/library/re.html). For example, rule-based syntax prevents the incorrect segmentation of "*B. lactucae* regulatory sequences in *P. infestans*" into "B. | lactucae regulatory sequences in P. | infestans." A dictionary of common potato pests and diseases was then used with SpaCy's pattern-matching functionality to expand the entity terms to enhance named entity recognition (https://spacy.io/). All sentences not containing entity information were removed, leaving a total of 21,995 sentences containing at least biological two entities.

## Named entity recognition

Knowledge extraction, a crucial aspect of knowledge base construction, encompasses both named entity recognition (NER) and relationship extraction (RE). NER in the field of biology primarily targets genes, diseases, chemicals, species, and mutations. PubTator Central (PTC; *Wei et al. (2019)*) was used in this study for biological entity recognition (*e.g.*, genes, diseases, chemicals, species, mutations, and cell lines). PTC, a web-based system, offers a multi-entity recognition interface—PubTator Restful API (https://www.ncbi.nlm.nih.gov/research/pubtator/api.html)—to export annotations. The named entity recognition annotations are displayed in Table 2. PMID denotes PubMed's unique identifier, "start" and "end" indicate the boundaries of mentions in the abstract, "mention" refers to the biological entity, "category" specifies the type of biological entity, and "standard" refers to standardized identification information of the biological entities.

In PubTator, diseases are identified based on Disease Ontology (*Schriml et al., 2012*), which is a database of human diseases. Because of this, PubTator's accuracy in recognizing crop-related diseases is relatively low. To address this issue, we enhanced disease and pest entity recognition using dictionary-based and pattern-matching methods, implemented through SpaCy's entity recognition tool (https://spacy.io/). SpaCy is a free, open-source library for advanced NLP in Python.

PubTator classifies pathogen information for potatoes under "species." To provide a more detailed representation of potato diseases and pests, we further categorized potato pathogens into fungal pathogens, bacterial pathogens, viral pathogens, and pests. Additional entities, such as abiotic stress and chromosome, were incorporated as needed to enrich the relationships within PotatoG-DKB. Ultimately, we identified data for 22 categories of entities. The detailed information on these categories is shown in Table 3.

Entities are represented in various forms in the literature, including abbreviations, word variants, and synonyms. For instance, "*Phytophthora infestans*" can also be referred to as "*P. infestans*", "*Phytophothora*", "*Pi*", or "*Phytophthora infestans* (Mont.) de Bary". Therefore, unique identifiers had to be established for each entity. Unique identifiers for entities such as genes, proteins, chemicals, and mutations were sourced from PubTator, while identifiers for species, fungi, bacteria, viruses, and pests were obtained from the Taxonomy database (https://www.ncbi.nlm.nih.gov/taxonomy). Entities without unique

**Table 2 Examples of entity annotation using PubTator.**

| PMID | Start | End | Mention | Category | Standard |
|---|---|---|---|---|---|
| 1351246 | 51 | 73 | *Phytophthora infestans* | Species | 4787 |
| 1351246 | 77 | 83 | Potato | Species | 4113 |
| 25681825 | 623 | 649 | Potato late blight disease | Disease | MESH:D055750 |
| 2132026 | 975 | 979 | Prp1 | Gene | 100795066 |
| 1655113 | 860 | 870 | Asparagine | Chemical | MESH:D001216 |
| 25873665 | 1050 | 1055 | V314I | Mutation | tmVar:p|SUB|V|314|I;HGVS:p.V314I;VariantGroup:1 |

**Table 3 Brief definition and examples for each of the 22 categories in the entity annotation.**

| S.No. | Annotation category | Brief definition | Examples |
|---|---|---|---|
| 1 | Disease | Diseases caused by potato infection pathogens | Late blight, soft rot |
| 2 | Fungus | Potato fungi pathogenic | *Phytophthora infestans* |
| 3 | Bacteria | Potato bacterial pathogens | *Pseudomonas syringae* |
| 4 | Virus | Potato virus pathogen | Potato virus Y, tomato mosaic virus |
| 5 | Pest | Potato pest pathogen | Potato cyst nematode |
| 6 | Species | Categorize of taxonomy database | Potato, tomato |
| 7 | Gene | Name of genetic factor | STWRKY8, 3-hydroxy-3-methylglutaryl-coenzyme a reductase |
| 8 | Mutation | Changes in nucleotide sequence of organism, virus or chromosome DNA genome | PiGPA1-deficient mutants |
| 9 | Chemical | Compounds involved in molecular biology research | Salicylic acid |
| 10 | Abiotic stress | The adverse effects of non-biological factors in the environment on potato plants | Drought, cold, heat |
| 11 | Biotic stress | The general term for biological stress. | Biotic stress |
| 12 | Protein | Protein, elicitin, enzyme | PR1, mitogen-activated protein kinase |
| 13 | Chromosome | Potato chromosome | Chromosome 8 |
| 14 | Organ | Organ of plant | Leaf, stem, tuber |
| 15 | Tissue | Type of cell or part of cell | Cytoplasm |
| 16 | Phenotype | The observable traits or characteristics of an organism | Violet flower colour |
| 17 | Structure | The structure of proteins | 1,134 amino acid residues |
| 18 | Trait | Measurable features | Early maturity, late blight resistance |
| 19 | mRNA | Messenger RNA | Star mRNA |
| 20 | Location | Pest and disease occurrence location | Mexico |
| 21 | QTL | Quantitative trait locus | GpaVvrn |
| 22 | Molecular biology tool | Equipment or software used in molecular biology experiments and research | RSOLA |

identifiers are represented as '–'. A potato-related entity dictionary was then constructed based on these entities and their categories. Examples from the entity dictionary are shown in Table 4, where "mention" indicates a meaningful item identified in potato-related

**Table 4 The examples of entity dictionary.**

| Mention | Category | Unique identifier |
|---|---|---|
| *Phytophthora infestans* | Species | 4787 |
| Potato | Species | 4113 |
| Potato late blight | Disease | – |
| Prp1 | Gene | 100795066 |
| Asparagine | Chemical | MESH:D001216 |
| *Erwinia carotovora* | Bacteria | 554 |
| V314I | Mutation | tmVar:p\|SUB\|V\|314\|I;HGVS:p.V314I;VariantGroup:1 |
| Potato psyllid | Pest | 290155 |
| *Alternaria solani* | Fungus | 48100 |
| Tomato leaf curl New Delhi virus | Virus | 223347 |
| Tuber | Organ | – |
| Low temperature stress | Abiotic stress | – |

**Table 5 LLM for relation extraction.** During the named entity recognition stage, we incorporate the identified entities into the prompt template as input for the LLM, enabling us to retrieve corresponding relationship information.

| Original sentence | Recognized entity information |
|---|---|
| Late blight in potato is caused by the fungus *Phytophthora infestans* and can inflict severe damage on the potato crop | ['late blight', 'potato', '*Phytophthora infestans*'] |

**Prompt**

In the sentence "Late blight in potato is caused by the fungus *Phytophthora infestans* and can inflict severe damage on the potato crop ", we have obtained the entities of ['late blight', 'potato', '*Phytophthora infestans*']. Please provide the relationship formed by these entities in the format of {subject, relationship, object}

**LLM output**

{"Late blight", "is a disease of", "potato"}
{"*Phytophthora infestans*", "causes", ,"late blight"}
{"*Phytophthora infestans*", "can inflict damage on", "potato crop"}

literature, "category" refers to the type of entity, and "unique identifier" is the unique identifier of the entity in the NCBI database.

## Relation extraction

Relation extraction (RE) is a subdomain of syntactic analysis aimed at revealing the semantic relationships between entities in unstructured text. In natural language processing, RE is typically treated as a supervised classification task. To achieve high accuracy in relationship extraction, a large amount of training data is required. Unfortunately, there is currently no specialized biological text *corpus* specific to potatoes. With the introduction of ChatGPT (*OpenAI, 2022*), large language models based on generative frameworks have shown promising performance across various tasks. In this study, LLM was used to extract triplet relationships. To enhance the efficiency of relationship extraction, we retained sentences containing at least two recognized entities.

**Figure 2 The potato ontology curation management platform.**

By constructing feature templates, we extracted the relevant relationships between entities using the Spark Language Model V3.5, a large language model released by iFlytek (https://xinghuo.xfyun.cn/sparkapi). An example of relationship extraction using this large language model is shown in Table 5. The extracted relationships, PMID, and sentences were then stored in a MySQL database.

We constructed a potato ontology by combining entity dictionaries and relational data. It is widely acknowledged that the co-occurrence of biological entities in a sentence indicates a strong correlation. LLM can accurately extract triplet information for highly correlated entities. However, their performance is poorer for entities with weaker correlations, resulting in incomplete relationship extraction. We developed a potato ontology platform to correct erroneous results, remove irrelevant data, and manually add relationships for those unrecognized. The potato ontology curation platform is illustrated in Fig. 2. This platform resulted in a more refined potato knowledge base, which includes 5,701 relational tuples. The species, fungi, bacteria, viruses, and pests were also expanded in the potato knowledge base using taxonomy dictionaries (https://www.ncbi.nlm.nih.gov/taxonomy), resulting in the addition of 5,003 new relationships and final total of 10,704 relationship data points. The relationship data between entities is shown in Table 6.

## Knowledge storage and visualization platform

Neo4j is an open-source graph database that stores structured knowledge in the form of graphs rather than traditional tables and supports semantic queries

**Table 6  The examples of relationship between entities.** In the relationship table, the data in the first section is obtained through LLM, while the data in the second section is expanded using the Taxonomy dictionary.

| Mention1 | Relation | Mention2 |
|---|---|---|
| Prp1 | Resistance to | *Phytophthora infestans* |
| *Ralstonia solanacearum* | Cause | Wilt |
| Snakin-1 | Gene in | Potato |
| Early blight QTL | On | Chromosome 2 |
| Late blight | Caused by | *Phytophthora infestans* |
| *Stenotrophomonas maltophilia* | Resistance to | *Ralstonia solanacearum* |
| Candidatus Liberibacter psyllaurous | Cause | Zebra chip |
| Quinone outside inhibitor (QoI) fungicides | Control | Brown spot |
| *Gibberella pulicaris* | Synonym | Fusarium roseum |
| *Gibberella pulicaris* | Authority | Fusarium roseum link, 1809 |
| *Gibberella pulicaris* | Scientific name | Fusarium sambucinum |
| *Gibberella pulicaris* | Authority | *Gibberella pulicaris* (Kunze) Sacc., 1877 |

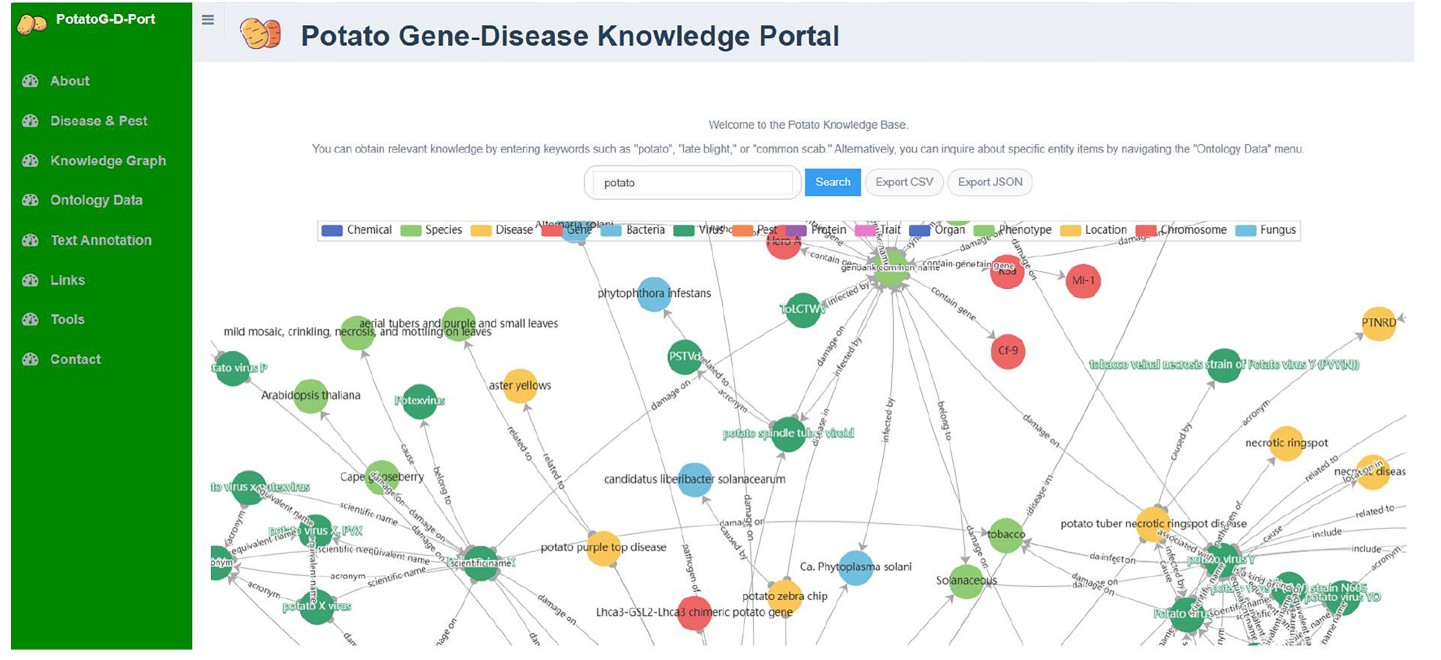

**Figure 3  The potato gene-disease knowledge graph visualization platform.**

(*Holzschuher & Peinl, 2013*). We chose Neo4j as the graph database for PotatoG-DKB. The Potato Gene-Disease Knowledge Portal (PotatoG-DKP) was developed using Django as the front-end interface, a high-level Python web framework, with Neo4j as the graph database and a Neo4j + MySQL combination as the back-end implementation. The PotatoG-DKP supports visual query, as shown in Fig. 3. Users can input an entity of

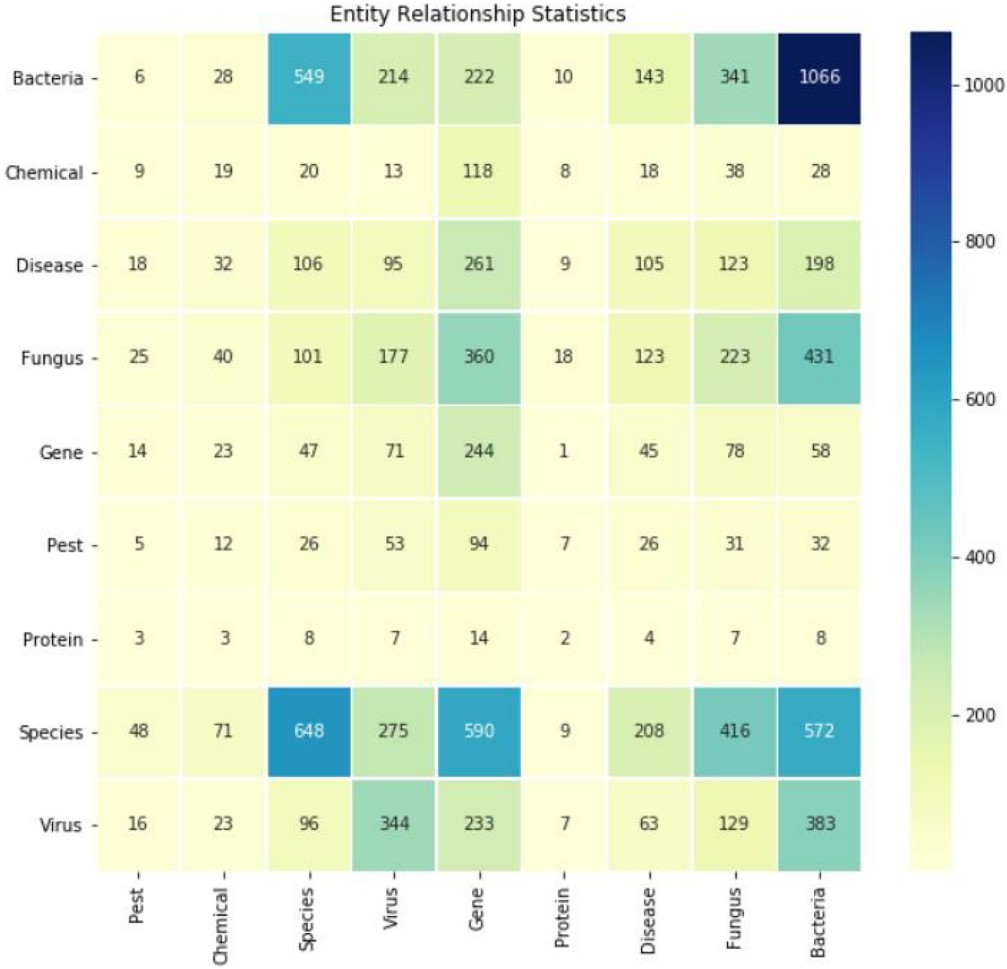

**Figure 4** Heatmap of several typical relationships extracted (*e.g.*, gene-disease, gene-gene, bacteria-bacteria).

interest to query related entities and relationships. A recursive method is used to support querying extended nodes and queried data can be exported in CSV (Comma-Separated Values) and JSON (JavaScript Object Notation) formats. If users do not input any query information, the portal can display the entirety of PotatoG-DKG. To ensure data security, a full data download option is not provided, but researchers can contact us for more information if they need access to the complete data.

## RESULTS

### Named entity recognition and relationship extraction

PubTator was used for named entity recognition and the Spacy tool was used for entity identification. Using the taxonomy dictionary to expand entities, we obtained a total of 5,206 entity items, including 300 genes, 205 diseases, 300 fungi, 300 bacteria, 109 pests, 300 viruses, 300 species, 277 chemicals, and 73 proteins as well as other entities.
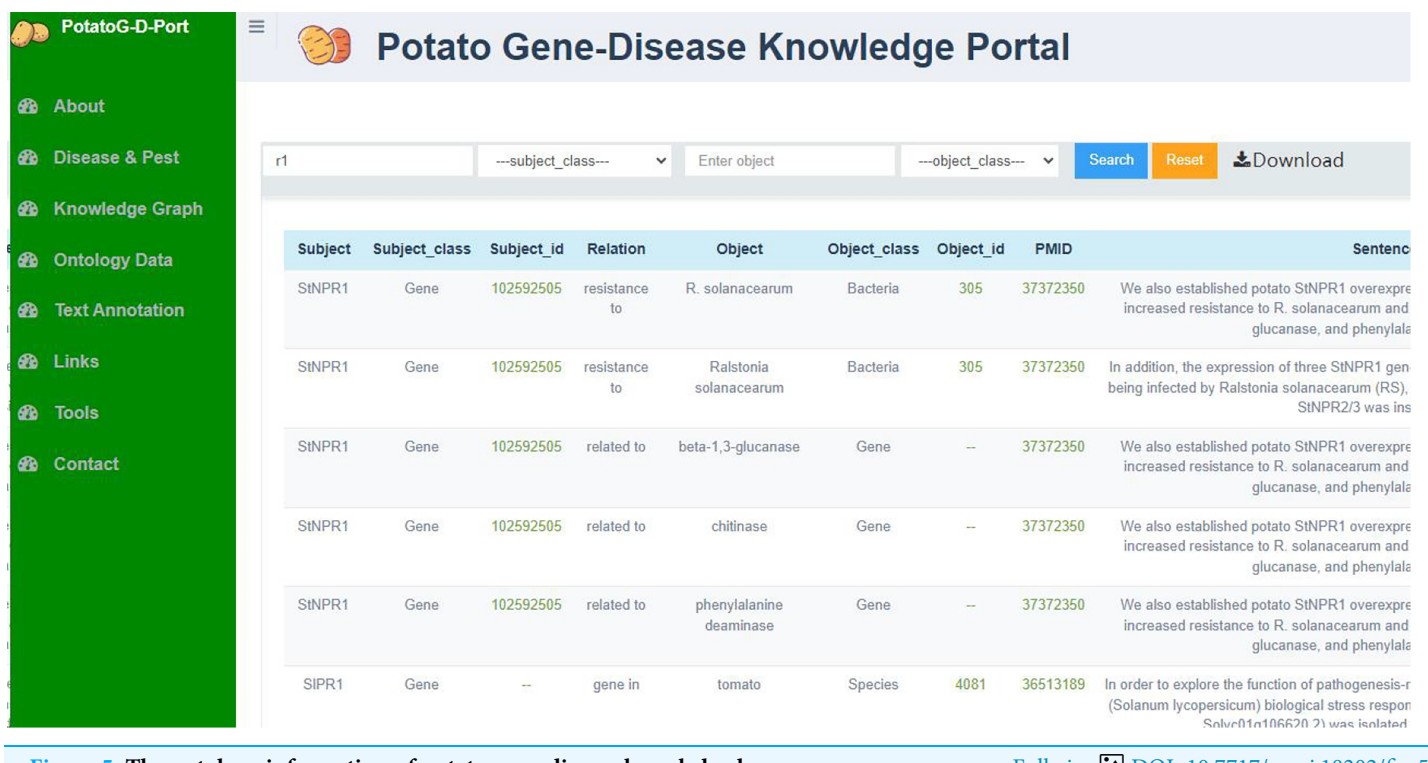

**Figure 5 The ontology information of potato gene-disease knowledge base.**

The Spark Large Language Model was used for relationship extraction, supplemented by manual curation. This process yielded a total of 10,704 relationship tuples. Figure 4 depicts the heatmap of typical relationships such as "gene-disease" and "fungus-disease." Each row and column respectively represent a category of interest (*e.g.*, genes, diseases, species), and each cell in the matrix represents the strength or frequency of the relationship between the two categories. The darker the color, the stronger the correlation between the two entities.

## Potato gene-disease knowledge base

Through the potato ontology data curation platform, we obtained potato ontology data on gene-disease relationships in potato as well as other relationships between biological entities, which, together, constitute the PotatoG-DKB, as shown in Fig. 5. This knowledge base includes entities, entity_categories, entity_id, relationships, PMIDs, and sentence information describing relevant relationships. Users can browse, search, and download relevant data. Additionally, users can retrieve information based on entity items or entity categories according to their needs. The PotatoG-DKB also provides an information tracing function, allowing users to link to the original PubMed literature through PMIDs or query detailed information about entities in NCBI using entity IDs. More detailed information can be accessed and downloaded from https://www.potatogd.com.cn.

## Potato gene-disease knowledge graph

The PotatoG-DKG covers 5,206 nodes and 9,443 edges. To make scientific literature directly and readily accessible to biologists and potato breeders, we developed a

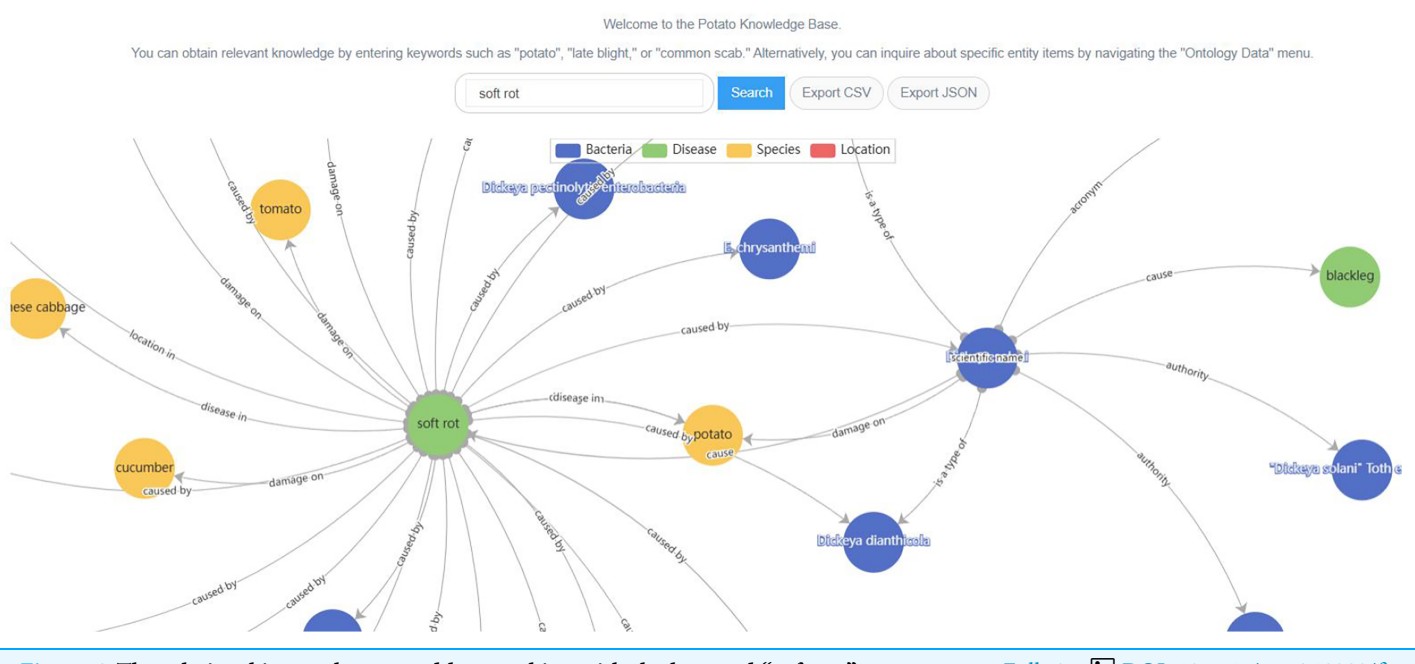

**Figure 6** The relationship graph returned by searching with the keyword "soft rot".

visualization platform, depicted in Fig. 3. The search function of the Neo4j graph database directly returns relevant relationships. For example, when searching for "A", it may return "A->B". In order to expand the retrieved relationships, a recursive search algorithm was adopted with a recursion depth limit of five to prevent excessive search depth and subsequent inefficiencies in querying. Figure 6 demonstrates the search results returned when using the keyword "soft rot" to search for relevant knowledge. A data download function is also provided, allowing users to export the data in both CSV and JSON formats.

## Potato gene-disease knowledge portal

Relevant researchers and breeders can access and query potato gene-disease ontology data through the PotatoG-DKP. As shown in Fig. 7, the "about" section of the portal website introduces the main process of potato gene-disease knowledge base construction. The "disease and pest" section displays common potato diseases caused by fungi, bacteria, viruses, and pests (Fig. 7A). The "knowledge graph" module enables visual querying of relevant information in the ontology database. To facilitate user operations, PotatoG-DKP has a fuzzy search function, and users can also download the entity-relationship information of interest (Fig. 7B). In the "ontology" section, PotatoG-DKP provides potato gene-disease ontology data, where users can also query or download relevant data (Fig. 7C). The "text annotation" module offers text annotation, allowing users to search and read abstracts of relevant literature (Fig. 7D). If they wish to delve deeper into the

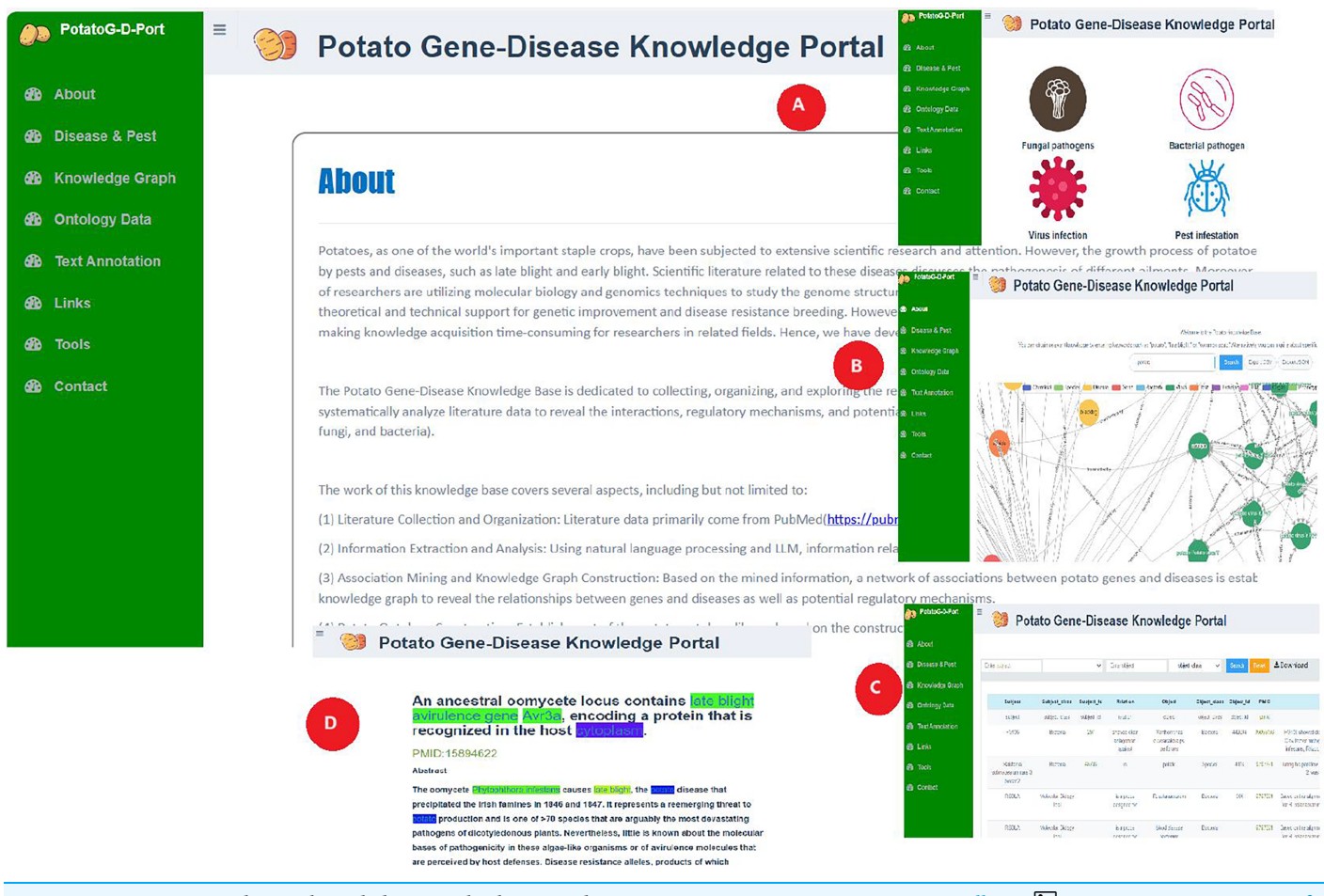

**Figure 7** Potato gene-disease knowledge portal schematic diagram.

specific content, they can directly access the original PubMed article through the PMID. "Links" to related databases are also provided, and users can access or download the tools used in this article through the "tools" section.

## DISCUSSION

The Potato Gene-Disease Knowledge Graph visualizes the network of related entities in a graphical manner, aiding in the discovery of hidden relationships between knowledge elements. Figure 8 illustrates the results obtained from searching for the gene entity named "Rx2" within the knowledge graph. These results include connections such as {"Rx2", "is similar to", "Rx1"}, {"Rx2", "is similar to", "Rxh1"}, {"Rxh1", "encodes", "Gpa2"}, and {"Gpa2", "resistance to", "*Globodera rostochiensis*"}. Although "Rx1" is not directly linked to "*Globodera rostochiensis*", "Rx1" may also confer resistance to this pathogen (as indicated by the green dashed lines in the figure); the relationship between "Rx2" and "*Globodera rostochiensis*" warrants further investigation. More information is available at https://www.potatogd.com.cn.

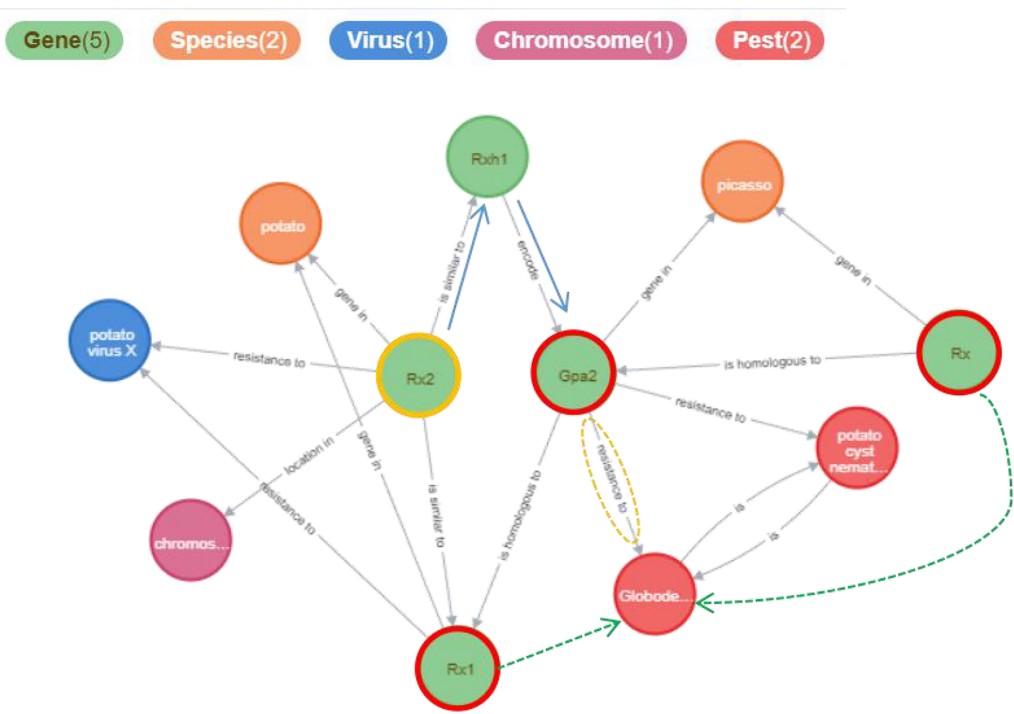

**Figure 8 The results returned from searching for the "Rx2" relationship using keywords in PotatoG-DKG.** We presented the relevant relationships of genes in the figure, with the green dashed lines highlighting relationships that are worthy of further investigation by researchers.

## CONCLUSIONS

This study used natural language processing technology and LLMs to analyze literature pertaining to potato gene-disease relationships. By performing entity recognition and relationship extraction, we constructed a knowledge graph comprising 5,206 nodes and 9,443 edges. To facilitate researchers in the use of this information, we developed a potato gene-disease knowledge portal (PotatoG-DKP) capable of effectively organizing and visually presenting the associated relationships. The findings of this study suggest that leveraging associations extracted from scientific literature can assist in uncovering hidden relationships among entities like genes and diseases, and integrating such data may foster the discovery of new hypotheses.

We will continuously update the Potato Gene-Disease Knowledge Base as new research articles are published, ensuring the data remains current and comprehensive. The raw data we used were article abstracts from PubMed, which reflect the key information of the article, but do not include the entirety of the information contained in the article. Therefore, in the future, we will consider processing the PubMed Central full-text database. By integrating multi-source data, we aim to enrich the knowledge base. Additionally, we plan to develop an intelligent question-answering assistant for potato gene-disease relationships based on the constructed PotatoG-DKB and LLM, further aiding potato researchers.

## ACKNOWLEDGEMENTS

We would like to thank Professor Zhongren Yang, a potato expert from the Inner Mongolia Agricultural University and Vice President of National Innovation Alliance of Northeast Mountain Wild Vegetable Industry, and his team for their help and guidance in data collation and review. We would also like to thank Professor Wenguang Zhang, a bioinformatics specialist of the College of Life Science, Inner Mongolia Agricultural University, for his guidance in data review.

### Funding

This work was supported by the Inner Mongolia Science and Technology Major Special Projects [2021ZD0005]; Natural Science Foundation of Inner Mongolia Autonomous Region [2020MS06013]; Inner Mongolia Agricultural University [JC2019005]; Basic Scientific Research Business Project of Universities in Inner Mongolia Autonomous Region [BR221022]; Inner Mongolia Agricultural University College Students' Innovation and Entrepreneurship Training Program [202210129003]. The funders had no role in study design, data collection and analysis, decision to publish, or preparation of the manuscript.

### Grant Disclosures

The following grant information was disclosed by the authors:
Inner Mongolia Science and Technology Major Special Projects: 2021ZD0005.
Natural Science Foundation of Inner Mongolia Autonomous Region: 2020MS06013.
Inner Mongolia Agricultural University: JC2019005.
Basic Scientific Research Business Project of Universities in Inner Mongolia Autonomous Region: BR221022.
Inner Mongolia Agricultural University College Students' Innovation and Entrepreneurship Training Program: 202210129003.

### Competing Interests

The authors declare that they have no competing interests.

### Author Contributions

- Congjiao Xie conceived and designed the experiments, performed the experiments, analyzed the data, prepared figures and/or tables, authored or reviewed drafts of the article, and approved the final draft.
- Jing Gao conceived and designed the experiments, analyzed the data, authored or reviewed drafts of the article, and approved the final draft.
- Junjie Chen analyzed the data, authored or reviewed drafts of the article, and approved the final draft.
- Xuyang Zhao analyzed the data, prepared figures and/or tables, and approved the final draft.

## Data Availability

The data is available at the Potato Gene-Disease Knowledge Portal: https://potatogd. com.cn; and at Zenodo: Xie, C. (2024). potato gene-disease knowledge base [Data set]. Zenodo. https://doi.org/10.5281/zenodo.12706091.

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
