# Peer review of "PotatoG-DKB: a potato gene-disease knowledge base mined from biological literature"

_PeerJ, doi:10.7717/peerj.18202_

## Round 0.1 · original submission · Major Revisions

The manuscript got critical remarks from two reviewers. Overall, the manuscript can't be accepted in current form. Please reorganize the text and check all technical details (download format, secure protocols) to fit to the database standards. Take more time for revision, if necessary.

**Language Note:** The review process has identified that the English language must be improved. PeerJ can provide language editing services - please contact us at [email protected] for pricing (be sure to provide your manuscript number and title). Alternatively, you should make your own arrangements to improve the language quality and provide details in your response letter. – PeerJ Staff

Reviewer 1 ·

Basic reporting

The authors, using NLP and AI (LLM) created “PotatoKB”, a knowledge graph of entities and relationships associated with potato disease. While PotatoKB has potential, given the current state of both the resource webpage and the manuscript, it is not ready to be published. Perhaps PotatoKB can become a valuable resource in potato disease research (and even the approach applied to plant species beyond potato), and I encourage the authors to refine their resource, and manuscript.

The manuscript is not well written, the language need to be significantly improved for flow, clarity, and readability.

The literature review in the background is written very haphazardly and confusingly.

The table and figures are either not described or annotated, or are not particularly effective at communicating any information.

The methods are not sufficient, and especially code, supplemental data and intermediate results are not supplied, for example the dictionaries used in pre-processing and knowledge extraction, and the final ontology (only a screenshot of a subset is provided).

The resulting knowledge graph needs to be provided in a usable and interoperable format.

Experimental design

The methodology appears to not have been done very rigorously, for example entities in the knowledge graph appear to be duplicated due to case sensitivity (e.g. there is both bacteria and Bacteria; disease and Disease). Genes do not use standard gene identifiers.

The authors are unclear in what they mean by “ontology”. An ontology is a common controlled vocabulary, defining a set of terms and relational expressions. PotatobKB uses an ontology to represent information mined from literature, but is not an ontology itself, and should attempt to use a standard, already existing ontology (e.g. https://obofoundry.org/ontology/pso).

Validity of the findings

The manuscript is lacking results on the coverage of the knowledge graph.

The authors need to additionally provide proof of validity or usefulness of the resource (for example, a case study), and examples of how it can be used.

Additional comments

The webpage providing the resource is not usable at this point:
- URL address is an IP address, which is not user friendly and the wepage is not secure (https), which is a security issue and will be inaccessibility for may users with a firewall
- None of the pages are documented to guide new users, there is explanation, no contact or about pages
- "Knowledge graph" page:
- no auto complete or fuzzy searching in any of the the search bars, so a user is not able to find any entities unless they already know the exact text to search
- description of nodes on hover are not text wrapped, and extend beyond the limit of the page, making them unreadable
- csv download is not csv
- JSON is valid, but does not contain any of the visible relationships
- nodes do not contain any information besides their name and database label
- nodes and relationship should contain an understandable description and their origin (or alternatively contain a link to the same search in the “Ontology data sheet” page
- "Ontology data sheet" page
- why is it called “Ontology data sheet”?
- nicely detailed, however has the same issue with the search bar as the graph
- download results in an error

·

Basic reporting

The authors developed a literature-mined potato gene-disease knowledge base by automatic extraction of the information from scientific literature abstracts and manual checking. The information is available via a website.
The manuscript is well-written and clear and the information collated is a first step to facilitate access to a large body of knowledge on potato diseases and the genes involved.

I have several concerns that I engage the authors to consider:

1-I would suggest providing links to all the other potato databases, for example AHDB https://potatoes.ahdb.org.uk/knowledge-library/potato-disease-identification in the manuscript and the website (see below)

2-The name of the database refers to a larger context (potato knowledge) than its actual content (Potato gene-disease knowledge). I would suggest to change the name “PotatoKB” in the paper title into “Potato Gene-Disease KB”.

3-The authors should make clear in their abstract and conclusion that the automatic extraction was carried out on the abstract of the scientific papers and not on the full texts.

4-Portal:
a-Change the name “Potato knowledge portal” into “Potato gene-disease knowledge portal”
b- I invite the authors to provide a new tag, below the “text annotation” tag, with the links to all the other potato databases, for example AHDB https://potatoes.ahdb.org.uk/knowledge-library/potato-disease-identification
(both in the manuscript and the website).

-5-Knowledge graph:

- a-it is not possible to click on the nodes. The mouse-over tool allows visualizing some textual information related to the node but not the original reference (with a PMID, a DOI etc. that one could easily copy). From my point of view this is a severe limitation of the knowledge graph option provided. Users do not only search information but they also want a quick access to the source data that they will check by themself.

-b-Textual search: if one introduces a mistake in the query (or simply a space after the name queried) the interface remains “silent”. A sentence “check spelling” might by useful. Automatic completion based on controlled vocabulary would be useful.

c-Content:
c1-The query “phytophthora” does not provide the node “Phytophthora infestans” with “include” as edge. Only P. parasitica and P. palmivora appear with the edge “include”. Why ?
c2-The graph contains Phytophtora (without the second “h”) : how does the knowledge base deals with spelling errors in the original data ?
-c3-The nomenclature for naming species is not respected (e.g. p. mirabilis instead of P. mirabilis); please consider writing correctly the species names.
-c4-The query Phytophthora mirabilis (with a P upper case) does not work when phytophthora mirabilis (lower case) provides a graph. Both shold provide the same answer.
Query “p. mirabilis” (without the quotes) provides a graph with a single node (no edges); this query should provide the same answer as the search "phytophthora mirabilis".
-c5-“phytophthora mirabilis” query indicates it is a pathogen of mirabilis jalapa but the query mirabilis jalapa provides a graph with a single node. A relation “is infected by” was expected.

6-Ontology datasheet:
a-It would be practical if the result of the query could be sorted by alphabetical order or else by clicking on top of the column names.
b-What is the rational behind the order of appearance of the results for given search. Are they some weights affected to the results ?
c-boolean operators do not work in the search field. Would it be possible to add them ?

7-Table 1 : check spelling of Phytophthora infestans; check lines breaks.

8-Table 5: chinses => Chinese

Experimental design

no comment

Validity of the findings

no comment

Additional comments

no comment

---

## Round 0.2 · accepted · Accept

The reviewers have no critical remarks. However, reviewer #3 has some comments about possible updates to the database based on novel published data. Please add a phrase to the discussion regarding this point.

·

Basic reporting

The manuscript and the associated knowledge-base from Congjiao Xie et al. have been improved following the suggestions. I would suggest to add legends to the Figures (only titles are indicated in the manuscript), describing what is shown in the figure.

Experimental design

ok

Validity of the findings

ok

Additional comments

none

Reviewer 3 ·

Basic reporting

The authors of the article presented a knowledge base they had created that reflected the relationship between various genes and potato diseases using automated processing and analysis of information from abstracts of other scientific articles on this plant object. Such a resource is of great interest to researchers and can be useful at the stage of collecting information and planning experiment design. Most of the comments in the presented version of the article have already been noted by other reviewers.

Experimental design

1. It is not entirely clear from the text of the article how the knowledge base created by the authors records possible updates on the role of certain entities in different potato varieties against pathogens when new articles are published? This is especially true for the role of genes involved in regulating defense response systems. If, for example, two articles indicate opposite dynamics of transcription activity of the gene under study due to some differences in the experimental conditions, how will this be reflected in your database? As far as I can judge from the "ontology data" section on the resource website, the same objects are simply listed one after another with explanations in the "sentence" section. In my opinion, this should be reflected in the article. In addition, I would recommend that the authors indicate how often the information on the resource they created is updated.

2. Continuing with the previous point: it is not entirely clear - is the type of these relationships indicated on your map? Different diseases can affect the activity of different genes in different ways. If so, how is it possible to determine the role of a specific gene in the pathogenesis of a plant disease? By manually reviewing information on each mention of it? In the chapter "(2) Potato Gene-Disease Knowledge Base" it is said that (quote) "This knowledge base includes sentence information describing relevant relationships..." in "ontology data" section. However, the data on genes presented in their current form seem to me to be unsystematized, simply listed. I would recommend that the authors divide the "ontology data" section into additional subsections within which it would be possible to systematize repetitive data on genes and organisms.

Validity of the findings

no comment

Additional comments

no comment